# SELF-SUPERVISED CONTRASTIVE ZERO TO FEW-SHOT LEARNING FROM SMALL, LONG-TAILED TEXT DATA

## ABSTRACT

For natural language processing (NLP) 'text-to-text' tasks, prevailing approaches heavily rely on pretraining large self-supervised models on massive external data sources. However, this methodology is being critiqued for: exceptional compute and pretraining data requirements; diminishing returns on both large and small datasets; and evaluation settings that overestimate performance differences. The core belief behind current methodology, coined 'the bitter lesson' by R. Sutton, is that 'compute scale-up beats data and compute-efficient algorithms', neglecting that progress in compute hardware scale-up is based near entirely on the miniaturisation of resource consumption. We thus approach pretraining from a miniaturisation perspective, such as not to require massive external data sources and models, and avoid translations from continuous input embeddings to discrete labels. To minimise favourable evaluation, we examine learning on a challenging long-tailed, low-resource, multi-label text classification dataset with noisy, highly sparse labels and many rare concepts. To this end, we propose using a 'dataset-internal', *self-supervised contrastive autoencoding* approach for pretraining that enables marked improvements in zero-shot, few-shot and supervised learning performance; even under a challenging, otherwise avoided, low-resource scenario, without defaulting to large-scale external datasets as support training signals. Crucially, we find evidence that zero and few-shot learning markedly benefit from adding more 'dataset-internal', self-supervised training signals, e.g. when increasing self-supervised learning signals via large external sources is infeasible.

## 1 INTRODUCTION

The current prevailing approach to supervised and few-shot learning is to use self-supervised pretraining on large-scale 'task-external' data and then fine-tune on end-task labels. Recent studies have found that, thus far, this way of pretraining fails in low-resource settings (Yogatama et al., 2019; Şerbetci et al., 2020) and that reported performance improvements are caused in part by evaluation setups that are designed in line with the paradigm that "massive resources are pivotal" to improving language understanding (Linzen, 2020; Schick & Schütze, 2020a; Dodge et al., 2020; Brown et al., 2020) or computer vision (Chen et al., 2020). Despite these critiques, the underlying goal of *better initialisation of layer weights* is a core requirement of successful learning with neural networks, where self-supervised layer-wise pretraining (Bengio et al., 2006) was replaced by better layer initialisation (Glorot & Bengio, 2010), which was in turn replaced by pretraining on growing amounts of external data (Bojanowski et al., 2017; Devlin et al., 2019; Chen et al., 2020; Brown et al., 2020) – i.e. FastText, BERT, SIMCLR and GPT-3. The latter three approaches require massive compute and data resources, but enable marked learning improvements in few-shot (SIMCLR, GPT-3) or zero-shot (GPT-3) scenarios compared to models that have several orders of magnitude fewer parameters. There are efforts to reduce model size requirements for few and zero-shot adaptation by orders of magnitude (Schick & Schütze, 2020a;b; Plank & Rethmeier, 2019), with some being increasingly beneficial in scenarios with low input data ($X$), label resources ($Y$), and rare events in $X, Y$. Crucially, such approaches do not simply rely on more data, but on creating better initialised input features $X$. In contrast, approaches like SIMCLR or BERT (Chen et al., 2020; Devlin et al., 2019) use self-supervision via contrastive learning and input masking on large-scale datasets to create broader learning signals than supervision provides. SIMCLR is based on a metric learning approach called contrastive self-supervision – i.e. learning to distinguish (dis-)similar inputs using

generated, but weak supervision tasks. However, as Musgrave et al. (2020) find, "when evaluating old vs. recent metric learning approaches, while *controlling for data and model size*, newer methods only marginally improve over the classic contrastive formulation". Remarkably, Bansal et al. (2020) recently showed that adding broader self-supervision rather than increasing data size during large-scale pretraining can substantially boost few-shot performance.

Our central question is whether *increased (broader) pretraining self-supervision also boosts few and zero-shot performance using only small-scale, 'task-internal' data, instead of resorting to large-scale pretraining on orders of magnitude more 'task-external' data* – i.e. **Do we really need large datasets for pretraining or just more (broader) self-supervised learning signals?** To broaden small data self-supervision, we propose a *contrastive self-supervised objective based on label-embedding prediction*, where labels are expressed as word embeddings to learn their matching with an input text embedding. For contrastive learning, our method samples positive and negative word input tokens $X$ for self-supervised pretraining, zero and few-shot learning; and positive and negative classes $Y$ for few-shot to fully supervised fine-tuning. Thus, we propose a model architecture that unifies training from labels $Y$ and inputs $X$. To increase evaluation robustness, we compare models of the same parameter and data sizes as suggested by Musgrave et al. (2020), and evaluate on a challenging learning problem as suggested by Linzen (2020); Hooker (2020). Namely, we evaluate against a challenging low-resource, long-tailed, noisy multi-label data settings, where information is always limited, since the long tail grows with data size and because its modeling requires the majority of parameters (Hooker et al., 2020b). For robust evaluation, we use a typical training, development, test setup and first establish a solid, supervised baseline for many-class multi-label classification that is optimised with a set of generalisation techniques proposed by Jiang et al. (2020). For evaluation in supervised, few and zero-shot learning scenarios, we analyse and propose evaluation metric choices which are meaningful across all scenarios for broader performance comparisons.

Contributions: ① We provide a straight-forward method for *self-supervised contrastive label-embedding prediction* and ② evaluate it in a challenging, noisy long-tail, low-resource multi-label text prediction scenario. ③ We show that small-scale 'data-internal' pretraining (on 8-80MB of text) not only improves supervised performance, but also strongly boosts few and zero-shot learning by *increasing self-supervision amounts* for small data, rather than increasing data amounts via the standard large-scale external data pretraining approach.

## 2 RELATED WORK

Large to Web-scale data pretraining is at the core of state-of-the-art methods in computer vision (Chen et al., 2020) and language processing (Rogers et al., 2020; Brown et al., 2020). However, challenges and disadvantages are increasingly being discussed. **(i)** A requirement of *large-scale external text data resources* (Yogatama et al., 2019; Schick & Schütze, 2020a), **(ii)** an inability to pretrain recent architectures on small-scale data (Liu et al., 2020; Melis et al., 2020; Şerbetci et al., 2020), **(iii)** calls for more challenging evaluation tasks (Linzen, 2020; McCoy et al., 2019) and (iv) diminishing returns of pretraining on large supervised datasets (Wang et al., 2020b). To address issue **(iii)**, challenging evaluations on long-tail prediction (Chang et al., 2019), few-shot (Schick & Schütze, 2020a), or zero-shot (Brown et al., 2020), were recently shown to benefit from self-supervised pretraining, but to-date, *require massive, 'task-external', pretraining datasets*. (c) Remarkably, Bansal et al. (2020) showed that **for large 'data-external' pretraining, using more self-supervision, not more data, also boosts few-shot performance**. This finding inspired us to collect evidence towards a core question: "Do we need massive data (signals) or just more (diverse) self-supervised learning signals for pretraining?". We collect evidence by posing three research questions and propose solutions that require designing approaches for issues (i-iii) as follows. One, to address issue **(i)**, "can increasing self-supervision signals during 'data-internal' pretraining on small data, i.e. without large-scale 'data-external' pretraining, boost few and zero-shot performance"? Two, to address issue **(ii)**, "what pretraining objectives and models do we chose that work without large training data"? Three, to address issue **(iii)**, "within what challenging learning scenario should we evaluate while incorporating the now standard "any NLP task as a 'text-to-text' problem" paradigm (Raffel et al., 2020)"?

Fortunately, existing techniques can be extended to address these issues. For example, *supervised* label embedding prediction (pre-)training enables few and zero-shot learning of subsequent (unseen) supervised tasks. However, *this requires the first (pre-training) task to be supervised*, unlike recent large scale self-supervised pretraining methods. Large-scale, self-supervised pretraining and label embeddings can be combined (Chang et al., 2019) to fine-tune externally pretrained BERT models via label embedding prediction to boost long-tail task performance. However, BERTs' contextualized word embeddings did *not work* as label embeddings ELMOs' word embeddings had to be used (3.2), further increasing resource requirements. Even worse, when comparing language model pretraining on small text corpora, Transformers (Wang et al., 2020a) largely underperform CNNs and LSTMs (Merity et al., 2017). Fortunately, Liu et al. (2017) established that label-embedding prediction CNNs boost long-tail prediction, even without modern self-supervision or using large 'task-external' data pretraining. Further, Pappas & Henderson (2019); Zhang et al. (2018b) used *supervised* text label-embedding (pre-)training and investigated transfer to subsequent supervised tasks, though not under long-tail evaluation. Here, label embeddings are average word embeddings over label description words – i.e. label descriptions are required. The former added noise contrastive estimation (NCE) (Ma & Collins, 2018) via negative sampling of labels to zero-shot predict rare, unseen classes post supervised pretraining on seen classes. Later, Jiang et al. (2019) adapted the same idea for zero-shot image classification via supervised pretraining on pairs of 'source' images and 'source' text label descriptions. They reduced overfitting by additionally pretraining on pairs of 'source' image and most similar 'zero-shot target class' text descriptions – though this is not technically zero-shot learning because sensible target label text descriptions have to be provided, which when unknown (zero-shot), again leads to the long-tail issue. All these approaches are loosely based on Matching Networks by Vinyals et al. (2016) and add various training objectives.

**We thus combine the advantages of *self-supervised pretraining* for large data with supervised label embedding prediction for smaller data to propose a contrastive self-supervised pretraining via label-embedding prediction usable for small data pretraining**. We extend the supervised label embedding baseline method by Zhang et al. (2018b) and add four important changes. First, we combine label and word embedding look-up tables into one table, as this pushes input words and label(-words) to remain in a shared vector space during training, when predicting dense label(-word) embeddings from dense input word embeddings. This 'dense-to-dense' prediction of words to label(-words) follows the current "any NLP task as a 'text-to-text' prediction" paradigm (Raffel et al., 2020), but avoids constant dense-to-sparse translation into label distributions via a compute intensive softmax. Secondly, we thus use a noise contrastive estimation (NCE) objective (Ma & Collins, 2018), replacing softmax normalization with negative sampling of (supervision) labels. Combining NCE and label embeddings allows predicting arbitrarily large class set (long-tails) and unseen classes. While Pappas & Henderson (2019) used NCE for supervised label pretraining, we add self-supervised pseudo-label (word) pretraining. Because labels and input words occupy the same vector space, we can use pseudo-labels (words) for self-supervised pretraining by sampling positive words from a current text instance, and negative words from adjacent text instances within a mini-batch. Three, we chose to sample from within a batch to reduce reliance (training bias) on knowing or expecting future and past word or label distribution statistics for the whole dataset, since in a zero-shot evaluation scenario unseen label and input word statistics are unknown. This also adds subsequent learning flexibility because no statistics collection preprocessing is required. Fourth, we add k-max pooling as in the CNN long-tail research by Liu et al. (2017), because it helps during zero-shot learning.

Such label-embedding based self-supervised pretraining has multiple advantages. It does not require large or external resources as in (i). Its *small 'data-internal' self-supervised word pseudo label pretraining addresses issue (ii) and enables unsupervised zero-shot learning. It also markedly boosts few-shot performance without requiring task external supervised annotations* as in (i) or supervised embedding transfer as in Pappas & Henderson (2019); Zhang et al. (2018b); Jiang et al. (2019). Since label embeddings are a common long-tail prediction technique, which addresses issue (iii), it makes our approach *suitable for low-resource, long-tail learning without task external labels or large-scale annotated datasets*. Finally, label embedding NCE training allows for (dense) 'text-to-text' training, making it applicable to a variety of tasks. We demonstrate the benefits of such a self-supervised pretraining method and model for self-supervised zero-shot learning (*input $X$-efficiency*) §6.4 or few-shot learning (*label $Y$-efficiency*) §6.3.

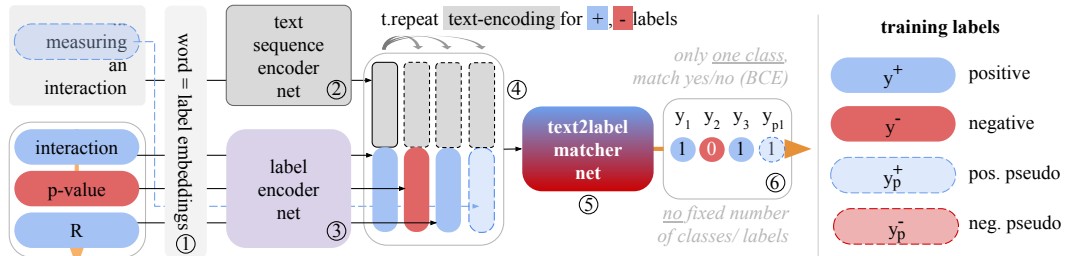

Figure 1: **Contrastive text-sequence-embedding-2-label-embedding matcher model:** A text ('measuring an interaction'), and positive ('interaction', R) or negative labels ('p-value') are encoded by the same word embedding layer $E$ ①, where labels have word IDs for lookup. The text embeddings are then encoded by a sequence encoder $T$ ②, while $c$ labels are encoded by a label encoder $L$ ③. Each text has multiple labels, so the text encoding $t_i$ is repeated for, and concatenated with, each label encoding $l_{i,l}^\circ$. The resulting batch of 'text-embedding, label-embedding' pairs $[[t_i, l_{i,1}^\circ], \ldots, [t_i, l_{i,c}^\circ]]$ ④ is fed into a 'matcher' classifier ⑤ that trains a binary cross entropy loss ⑥ on multiple (pseudo-)label (mis-)matches $\{0, 1\}$ for each text instance $t_i$, resulting in a noise contrastive estimation objective (NCE). Words like 'measuring' provide self-supervised pseudo-labels (left). Positive and negative (pseudo-)labels are sampled from their own or other instances in a minibatch. Unlike Zhang et al. (2018a) we use a CNN for ②, negative sampling and self-supervision.

## 3 SELF-SUPERVISED, CONTRASTIVE DENSE-TO-DENSE TEXT PREDICTION

In this section, we propose to use label-embeddings, previously used for supervised learning only (Pappas & Henderson, 2019; Zhang et al., 2018b), and exploit them for *self-supervised contrastive pretraining* on small-scale data. This enables contrastive self-supervised pretraining somewhat similar to methods used for large-scale models. However, we only use small-scale 'task-internal' data for pretraining, which requires orders of magnitude less data and compute than large-scale, 'task-external' pretraining approaches. Most NLP models translate back and forth between discrete words and continuous token embeddings, often using a softmax computation that is limited to predicting classes known at training time. To ease learning from small data, our *first core idea is that text input words $w_i \in x$ and labels $w_{i,l}^\circ$ should be mapped into the same word representation space, i.e. drawn from a shared embedding look-up table $E$, to replace dense to sparse translations with embedding-to-embedding matching. Thus turns NLP from a discrete 'text-to-text' tasks, as proposed in Raffel et al. (2020), into a 'dense(text)-to-dense(text)' task.* We thus replace learning instance labels $y_i$ by their corpus-internally pretrained FastText or randomly initialised word embeddings $l_i^\circ \in L$, while others (Pappas & Henderson, 2019) use text descriptions to form label embeddings as the vector average over description word embeddings. As a result, *pretraining word embeddings also pretrains (favourably initialising) label embeddings.* Unknown labels ( words), in turn, can be inferred via methods like FastText subword embeddings (Bojanowski et al., 2017).

As outlined visually, left to right in Fig. 1, learning multi-label classification then becomes a contrastive learning problem of *matching the word-sequence embedding $t_i$ of text $i$* ②, with its $c$ label (word-sequence) embeddings $l_i^\circ = \{l_{i,1}^\circ, \ldots l_{i,c}^\circ\}$ ③, by feeding $c$ text-vs-label combinations $[[t_i, l_{i,1}^\circ], \ldots, [t_i, l_{i,c}^\circ]]$ ④ to a binary classifier $M$ ⑤ for matching. This means that instead of predicting $c$ classes at once, we predict a batch of $c$, single-class, binary classifications using binary cross entropy ⑥, where $c$ needs not be constant across instances $i$. The details of steps ① to ⑥ are as follows. To train a binary classifier, we need both positive and negative labels. Thus, for each text instance $w_i = \{w_a, \ldots w_z\}$ we want to classify, we need $g$ positive labels $w_i^- = \{w_1^+, \ldots w_g^+\} \in R^g$ and $b$ negative labels $w_i^+ = \{w_1^-, \ldots w_b^-\} \in R^b$ to form a label selection vector $w_i^\circ = \{w^+ \oplus w^-\} \in \mathbb{R}^{g+b}$. To indicate positive and negative labels, we also need a $g$ sized vector of ones $\mathbf{1} \in \mathbb{R}^g$ and a $b$ sized zero vector $\mathbf{0} \in \mathbb{R}^b$, to get a class indicator $\mathbb{I}_i = \{\mathbf{1} \oplus \mathbf{0}\} \in \mathbb{R}^{c=g+b}$. Both the text (word) indices $w_i$ and the label indices $w_i^\circ$ are passed through a shared 'word-or-label embedding' look-up-table $E$ ①, after which they are passed through their respective encoder networks – $T$ as text-sequence encoder, $L$ as label encoder. Thus, the text-encoder produces a (single) text embedding vector $t_i = T(E(w_i))$ per text

instance $i$ ②. The label-encoder produces $c = g + n$ label embedding vectors ($l_i^\circ$) that form a label-embedding matrix $L_i = [l_1^+, \ldots, l_g^+, l_1^-, \ldots, l_b^-] \leftarrow L(E(w_i^\circ))$ ③. As text-encoder $T$ we use a (CNN→max-k-pooling→ReLU) sub-network, while the label-encoder $L$ is simply an (average-pool) operation, since a single label ($w_{i,j}^\circ$), e.g. 'multi'-'label', can consist of multiple words. To compare how similar the text-embedding $t_i$ is to each label-embedding $l_{i,j}^\circ$, we repeat $t_i$ $c$ times and combine text and label embeddings to get a text-vs-label-embedding matrix $M_i = [[l_{i,1}^+, t_i], \ldots, [l_{i,c}^-, t_i]]$ ④ that is passed into the matcher network $M$ ⑤ to produce a batch of $c$ probabilities $p_i = \{\sigma(M(M_i)_1), \ldots, \sigma(M(M_i)_c)\}$ ⑥. As the optimisation loss, we the use binary cross entropy (BCE) between $p_i$ and $\mathbb{I}_i$, i.e. $\frac{1}{c}\sum_{l=1}^{c} \mathbb{I}_{i,l} \cdot log(p_{i,l}) + (1 - \mathbb{I}_{i,l}) \cdot log(1 - p_{i,l})$. Summing BCE over positive and negative (pseudo-)labels is referred to as noise contrastive estimation, as used in representation learning methods across fields (Ma & Collins, 2018).

Via pseudo-label embedding pretraining, a model can predict supervised labels absent prior supervision. This exploits both *transfer learning from inputs and labels*, using the matcher as a learned similarity function. Positive labels $w_i^+$ can be supervision labels. Negative labels $w_i^-$ can be sampled from the positive labels of other instances $w_j^+$ in the same batch, *which avoids needing to know the label set beforehand*. Since *labels are words*, we can sample positive words from the current and negative words from other text instances to get pseudo-labels. *Sampling pseudo-labels provides a straight-forward contrastive, partial autoencoding mechanism usable as self-supervision in pretraining or as zero-shot learner*. Because both real and pseudo labels are sampled words, the model does not need to distinguish between them. Instead, learning is controlled by an out-of-model sampling routine for real supervision and pseudo self-supervision labels. This leads to a *second core idea: once inputs X and outputs Y are well initialised, the model $\Theta$ can also be better initialised by pretraining via self-supervision. As a result, we can learn supervised, few and zero-shot tasks in a unified manner*.

## 4    SMALL, LONG-TAILED DENSE-TO-DENSE TEXT (LABEL) PREDICTION

Since it is our goal to research *better zero and few-shot learning approaches for small 'text-to-text' pretraining models*, we choose a small multi-label question tag prediction dataset as a test bed. We use the "Questions from Cross Validated"[1] dataset, where machine learning concepts are tagged per question. This dataset fulfills three requirements: it is small-scale, long-tailed, and entails solving a challenging, noisy 'text-to-text' prediction task. *There is currently no published baseline for this task*. The classes (tags) and input words are highly long-tailed (imbalanced). The first 20% of labels occur in only 7 'head' classes. Tags are highly sparse – at most 4 out of 1315 tags are labelled per question. Word embeddings are pretrained with FastText – details in appendix App. A.4. We use the labelled questions part of the dataset, which has 85k questions and 244k labels. What makes this problem particularly challenging is that $80\%$ of the *least frequent labels* are distributed over $99.5\%$ of classes, as an extreme long tail. The label density (% of active labels per question) is only $0.22\%$ or $\approx 2.8/1305$ possible classes per instance. For a realistic evaluation setting, we split the dataset diachronically, using the 80% earliest documents for training, the next 10% for development, and the last 10% for testing.

**Why not large external pretraining?** Real-world, long-tailed datasets are always dominated by a low-learning-resource problem for most classes. This makes two things obvious: (A) that *model learning cannot simply be solved by using massive data sets as the long-tail problem grows as well*; (B) that *studying self-supervised pretraining on challenging,* but smaller, *long-tailed datasets such as this one, is useful for assessing an approach's ability to learn from complex, real-world data*. Massive data pretraining masks and thus prevents studying these effects. We thus evaluate the effects of self-supervision in a noisy low-resource setup, also as a response to recent critiques of the evaluation metrics used to assess Web-scale learning (Linzen, 2020; Yogatama et al., 2019). As McCoy et al. (2019) shows, these evaluation setups are solvable by large-scale pattern overfitting, which, they find, leads to a 'Clever Hans effect', rather than real task progress.

---

[1]https://www.kaggle.com/stackoverflow/statsquestions

## 5 EXPERIMENTAL SETUP AND METRICS

We want to analyse the *benefits of self-supervision for (a) fully supervised, (b) few and (c) zero-shot learning in a noisy low-resource, long-tailed, multi-label classification setting*. In this section, we describe suitable evaluation metrics, then discuss results in the next section.

**Long-tail evaluation metrics and challenges:** Long-tail, multi-label classification is challenging to evaluate. Many classification metrics are unsuitable for evaluating long-tailed datasets. They either: (i) misrepresent performance under class imbalance; (ii) do not scale to many classes; or (iii) are only meaningful if the desirable number of classes per instance is known (multi-label classification). For problem (i) $ROC_{AUC}$ is known to overestimate imbalanced performance (Davis & Goadrich, 2006; Fernández et al., 2018), e.g. $ROC_{AUC}$ test scores were upwards of .98 for most of our models. For problem (ii), measures such as F-score require discretisation threshold search for imbalanced prediction problems, i.e. searching for the optimal threshold per class (on a development set), which becomes computationally infeasible. Simply using a $0.5$ probability threshold drives model selection towards balanced prediction, mismatching the long-tail problem. Metrics like precision@k handle problem (i-ii), but require knowledge of $k$, i.e. problem (iii): these metrics can only compare a chosen number of labels $k$, and cannot handle cases where the correct number of labels per instance varies or is unknown (label distribution shift). To more reliably measure performance under imbalance (i), to avoid unscalable class decision thresholding (ii), and to not optimise models for a set number of labels $k$ per instance (iii), we use the average-precision ($AP$) score. It is defined as $AP = \sum_n (R_n - R_{n-1})P_n$, where $P_n$ and $R_n$ are the precision and recall at the $n$th threshold. $AP$ measures classifier performance over *all decision thresholds*, is computationally cheaper than threshold search, and allows for a dynamic number of labels per class. This latter property makes this task especially hard. A model has to learn when to predict a label, at what rarity, and how many such labels to predict for each instance. We also report the macro-averaged Brier-Score ($BS$) over all classes, as a scalable, compute-efficient measure of classifier calibration. Though more accurate measures exist, computing them is more involved and they require additional evaluation labour when optimising a specific supervised dataset, which is not our goal. For both measures, we use their popular scikit-learn implementations[2].

**A challenging task, even for humans:** On the dataset it is hard to guess how many labels per question to tag and how specific they should be, especially without domain knowledge. Out of the different weighting schemes for average precision, we choose $AP_{micro}$ and $AP_{macro}$, as they are the most pessimistic (hardest to increase) measures to reduce optimistic evaluation. This choice is motivated by the goal of this work, which is to not simply to push end-task performance, but to use supervised learning scores as a proxy to evaluate the effects of pretraining on zero-shot learning as well as data-efficiency and speed of supervised and few-shot learning.

## 6 RESULTS

In this section, we first analyse a normal and a strong supervised baseline to minimise favourable comparison against subsequently evaluated label-embedding and self-supervision enhanced approaches. Finally, we analyse the benefits of 'dataset-internal' pretraining for few-shot learning, and how the amount of pretraining learning signal and model size affect zero-shot learning. *Test scores are reported according to the best dev set average precision score $AP_{micro}$ over all classes.*

### 6.1 BASELINE MODEL RESULTS

In this section, we establish baseline results (**BASE**) for a non-learning majority class baseline (ZeroR), a common ('weak') CNN baseline trained with binary-cross-entropy, and a solid CNN baseline optimised using a set of generalisation techniques proposed by Jiang et al. (2020). The **ZeroR** classifier is useful for establishing a baseline performance under class imbalance – e.g. if a class is present in only 10% of instances, then 90% accuracy is achieved by simply always predicting zero – i.e. the majority class. When doing so on our long-tailed task, where the class majority is always zero, we get an $AP_{micro}$ and $AP_{macro}$ of $0.2\%$, since out of the 1315 classes, maximally four classes are active per instance. Importantly, this tells us that: (a) simply learning to predict zeros can not

---

[2]https://scikit-learn.org/stable/modules/model_evaluation.html

Table 1: **Supervised prediction results:** comparing an optimized baseline (OB) with contrastive methods (CM: 4-12). CMs compare training from scratch vs. pretrain→fine-tune vs. self-supervised pretraining for few and zero-shot learning. Given the same hyperparameters (*), all CMs reach similar supervised end-task performance, while self-supervised CMs produce *fundamentally different results for zero and few-shot learning* – see subsection details.

| Training method/ model | learning setup | AP micro/ macro test % | Brier score macro |
|---|---|---|---|
| **BASE:** *baselines, only supervised learning (SL)* | | | |
| (0) ZeroR | always predict majority per class (=all zero) | 00.20/00.20 | n.a. |
| (1) WB: weak baseline (BCE) | supervised | 33.75/n.a. | n.a. |
| (2) OB: optimized baseline (BCE) | supervised | 45.01/22.81 | 0.0015 |
| **FROM SCRATCH (s):** *train from scratch: supervised (SL), or self+supervised (S(+S)L) – no pretraining* | | | |
| (3) (*) SL+SSLscr: h-params base | supervised + self-supervised from scratch | 47.13/25.28 | 0.0028 |
| (4) SLscr: h-parms like (*) | supervised scratch | 47.74/26.05 | 0.0028 |
| **SSL-PRETRAINED (p) → FINE-TUNE (f):** *self-supervised (SSL) pretrain → then fine-tune (SL or SL+SSL)* | | | |
| (5) SSLpre→SL+SSLfin: h-parms like (*) | SSL pretrain >SL+SSL fine-tune | 48.20/25.58 | 0.0027 |
| (6) SSLpre→SLfin: h-parms like (*) | SSL pretrain >SL fine-tune | 47.53/25.65 | 0.0028 |
| **FEW-SHOT:** *few-shot 10% train, 'pretrained then fine-tuned' (pf) vs from scratch (s)* | | | |
| (7) SSLpre→SLfin: h-parms like (*) | self pretrain >10% supervised fine-tune | 38.01/18.31 | 0.0037 |
| (8) SSLpre→SL+SSLfin: h-parms like (*) | self pretrain >10% self+supervised fine-tune | 38.25/18.49 | 0.0038 |
| (9) SLscr: h-parms like (*) | 10% supervised from scratch | 30.46/13.07 | 0.0032 |
| (10) (*) SL+SSLscr: | 10% self+supervised from scratch | 30.53/13.28 | 0.0039 |
| **ZERO-SHOT:** *zero-shot, self-supervised pretrain only* | | | |
| (11) SSLpre→0: h-parms, like (*) | self pretrain >zero-shot | 10.26/10.70 | 0.1139 |
| (12) SSLpre→0: extra h-param tuning | self pretrain >zero-shot | 14.94/14.86 | 0.0791 |

score well on under this metric and (b) that this problem setting is challenging. Next, we evaluate both **a weak and optimised baselines (WB, OB).** When using a very small CNN as baseline (WB) with max pooling over 10 filters at filter sizes 1-3 that feed into a one-layer classifier, we achieved $33.75\% AP_{micro}$ on the test set – after only tuning the learning rate. When tuning this baseline for parameters known to increase generalisation using a set of such methods suggested by Jiang et al. (2020), we get a more solid test score of 45.01 $AP_{micro}$ and an of 22.81 $AP_{macro}$. The macro result tells us that not all classes perform equally well. Upon closer inspection, we find that model performance worsens with increasing class rarity as expected. While establishing a solid baseline, we find expected limitations of model width, max-k pooling and dropout scale-up, and a confirmation that controlled experiment comparisons that only change one variable at a time, do not suffice to find better hyperparameter configurations. For example, when widening lower layer components and observing a decrease in performance, higher layers should also be made wider to accommodate the additional feature information from lower layers – which is consistent with findings in Nakkiran et al. (2020). A more detailed breakdown of this analysis can be found in Table Tab. 2 in the appendix App. A. We explore a considerable amount of hyperparameter configurations in an effort to compute a solid baseline. This allows for more robust insights and helps to speed up optimisation of the self-supervised models.

## 6.2 100% SUPERVISION (SL+SSLSCR) AS REFERENCE (*) FOR FEW AND ZERO-SHOT (0%)

Tab. 1 show both: models trained FROM SCRATCH (s), and models that are first PRETRAINED (p) using self-supervised word pseudo-labels from text inputs, and afterwards fine-tuned (f) on supervision labels. To fit the supervised end-task (tag prediction), both fine-tuning and training from scratch can either: (4) only fit supervision labels (SL) or (3) jointly fit supervised labels and self-supervised word pseudo-labels (S(+S)L), as described in §3.

**However, before analysing results, we define a controlled experiment setup using a fixed, but shared hyperparameter setting '(*) SL+SSLscr' as a reference (*).** Since SL+SSLscr is the most basic model learning setup that uses both self-supervision and supervision, we use its optimal

hyperparameters '(*) SL+SSLscr' as a fixed reference configuration for most subsequent learning setups, as indicated by the 'params like (*)' marker. This ensures a more controlled comparison of the effects of pretraining vs. training from scratch, and robust insights on how to design self-supervision during end-task fitting and pretraining. **The (*) reference will hence be used for most few and zero-shot settings.** When comparing PRETRAINED models with models trained FROM SCRATCH, we see that *under comparable hyperparameters, without setting-specific parameter tuning, all four learning setups perform similarly* within 1 percent point (%p) of each other. We also see that the PRETRAINED (5) model which uses self-supervision during both pretraining and fine-tuning performs best. Training FROM SCRATCH using self+supervision SL+SSLscr somewhat hurts performance compared to using supervision alone in SLscr. Test scores are reported for the best dev set $AP_{micro}$ scores.

### 6.3 FEW-SHOT: PRETRAIN FOR BETTER LONG-TAIL, LOW-RESOURCE, FEW-SHOT LEARNING

In this section, we present evidence that even in a data-limited, long-tailed setting, self-supervised *'data-internal'* pretraining: (a) increases few-shot learning performance of subsequent fine-tuning, while (b) improving learning speed and stability. This demonstrates that small data pretraining has similar benefits as large-scale pretraining (Brown et al., 2020; Schick & Schütze, 2020a). In

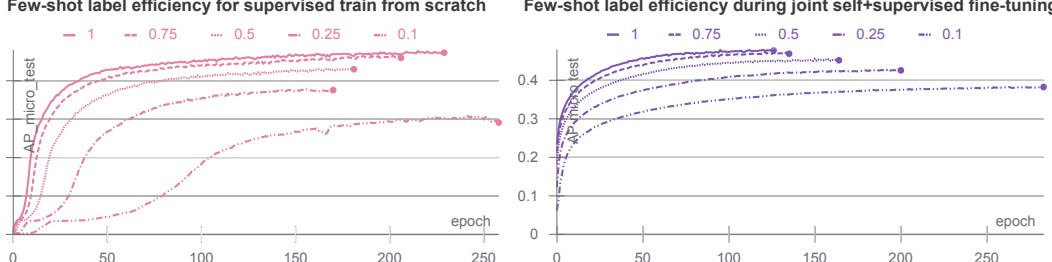

Figure 2: **Few-shot learning: Best training from scratch (left) vs. best fine-tuned (right)**: $AP_{micro\_test}$ curves for different few-shot portions: 100%, 75%, 50%, 25%, and 10% of training samples. *'Dataset-internal' pretraining via self-supervision (right) markedly improves few-shot learning performance, speed and stability* compared to training from scratch (left).

Fig. 2, when using the (*) reference model from Tab. 1, we now compare training from scratch (4) as before (pretraining off, left), with pretraining via self-supervised word pseudo-labels, and then fine-tuning on the supervised training labels (5) of the end-task (pretraining on). Note that our model architecture (Fig. 1) does not distinguish between self-supervised and supervised labels, which means that during self-supervised pretraining, we sample as many word pseudo-labels as real labels during supervised fine-tuning (or when supervising from scratch).

When fine-tuning the pretrained model on an increasingly difficult FEW-SHOT portion of (100%), 75%, 50%, 25% and only 10% of the supervised training data, we see large $AP_{micro|macro\_test}$ performance improvements compared to training FROM SCRATCH in both Tab. 1 and Fig. 2. On the right, in Fig. 2, we see that the pretrained models start with a higher epoch-0 performance, train faster, are more stable and achieve a markedly better few-shot end performance than the left-hand 'from scratch' setting. This is confirmed by detailed results for the 10% FEW-SHOT setting in Tab. 1, where pretrained models (SSLpre→SLfin, SSLpre→SL+SSLfin) achieve $\approx .38/.18 AP_{micro|macro\_test}$ compared to only $\approx .30/.13 AP_{micro|macro\_test}$ for models trained from scratch (compare (7-10). This means that, *when using only 10% supervised labels, pretrained models still retain* $38.25/48.20$, *or roughly* $80\%$, *of their fully supervised performance*. This provides evidence to answer the underlying question: "Do we really need more data for pretraining or can we simply increase self-supervision?". Very recent work by Bansal et al. (2020) has investigated this question for large-scale, self-supervised pretraining, where they showed that increasing self-supervision to create "a richer learning signal" benefits few-shot performance of large models. *Our results demonstrate that this is also the case for small-scale, non-Transformer pretrained models, even under a much more challenging long-tailed learning setting* than Bansal et al. (2020) examined. However, to better understand the benefits of using more self-supervised training signals

Figure 3: **Zero-shot performance by model and signal size: Left plot:** When using the same label and parameter amount as for the 'joint self+supervised train from scratch' reference model (*), allowing more self-supervision labels (left middle curve) and widening the network (left top curve) noticeably boosts zero-shot performance (supervised $AP_{micro\_dev\ and\ test}$). **Right:** when using less training data text (few-shot on inputs $X$), zero-shot still works, but we need to *wait much longer*.

and its relation to model size, we examine the zero-shot performance of our pretraining approach in regards to *label (signal) amount, network width and zero-shot $X$ data-efficiency* (low-resource zero-shot performance) – i.e. zero-shot performance when pretraining on fractions of inputs $X$ to forcibly limit self-supervision.

## 6.4 ZERO-SHOT: MORE IS BETTER, FOR 'LOW-RESOURCE' ZERO-SHOT PRETRAIN LONGER

In this experiment, we study how the number of self-supervised labels (signal) and the model width used for self-supervised pretraining affects zero-shot performance on the end-task test set. We show results in both Fig. 2 (11, 12) and Tab. 1 (ZERO-SHOT). In Fig. 2, we see that when using the reference hyperparameter configuration ((*) in Tab. 1), pretraining gets the lowest zero-shot performance. When increasing the number of self-supervised word pseudo-labels from 150 to 500, the model performs better (middle curve), while not using more parameters – so *increasing self-supervision signals is beneficial*. When additionally tripling the network's sequence and label encoder width and doubling the label match classifier size, zero-shot performance increases even more (top curve). This indicates that *for zero-shot learning performance from pretraining, both the amount of training signals and model size have a significant impact*. While increased model size has been linked to increased zero-shot performance of Web-scale pretrained models like GPT-3 (Brown et al., 2020), the influence of signal amount on zero-shot learning is much less well understood, because large-scale pretraining research often increases training data size when changing self-supervision, as outlined by Liu et al. (2020). Finally, in Fig. 3 we see that when pretraining our model for zero-shot prediction on only portions (100%, 75%, .50%, 25% and 10%) of the training text inputs $X$, i.e. an increasingly low-resource zero-shot setting, we still converge towards comparable full zero-shot performance (if we had not stopped early). However, each reduction in training size multiplies the required training time – when using the same number of self-labels. *This provides a promising insight into self-supervised pretraining on small datasets, which, if designed appropriately, can be used to pretrain well-initialised models for supervised fine-tuning and few-shot learning from very small text sizes.*

## 7 CONCLUSION

We showed that label-embedding prediction, modified for self-supervised pretraining on a challenging long-tail, low-resource dataset substantially improves low-resource few and zero-shot performance. We find that increased self-supervision, in place of increased data size or resorting to large-scale pretraining, strongly boosts few and zero-shot performance, even in challenging settings. In future, we envision that the proposed methods could be applied in scenarios where little in-domain (pre-)training data is available, e.g. in medicine (Şerbetci et al., 2020), and where new labels rapidly emerge at test time, e.g. for hashtag prediction (Ma et al., 2014). The code and data splits will be published on `https://github.com`.

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

# A  APPENDIX

## A.1  A BASELINE TUNED USING GENERALISATION TECHNIQUES

Table 2: **Building an optimised supervised baseline:** using test set generalization techniques as proposed by Jiang et al. (2020). %p denotes absolute percent points. Since parameters cannot be tuned in isolation, %p only reflects drops by deviating from optimal settings once they are found. Details on the explored hyperparameters are found in Tab. 3.

| Model | variable | observation | optimal parameter, %p drop from not using it |
|---|---|---|---|
| pre-opt NN | learning rate optimized | base setting | $33.75\%p\ AP_{micro\_test}$ |
| optimized NN | Optimal parameters ↓ | base setting | $45.01\%p\ AP_{micro\_test}$, $.0015\ BS_{macro}$ |
| larger NN | max-k pooling | important | max-3 pooling, 3%p better than max-1 pooling |
| | CNN filter size | important | n-gram filter sizes >2 matter ($\sim$2%p), comparing same filter amounts |
| | num CNN filters | important | 100 filters per n-gram size |
| | wider classifier | overfitting | more than a 1 layer classifier lead to overfitting |
| dropout | on CNN output | improvement | 2% better $AP_{micro\_test}$ test, 2%p improvement |
| | on deeper/ wider clf | none, stability | stabilizes learning, but same performance |
| optimizer | ADABOUND | failed | -39%p drop $AP_{micro\_test}$, despite tuning |
| learning rate | lower LR | crucial | LR = 0.0075 for ADAM with cross-entropy |
| batch size | batch size | important | batch_size = 1024 worked well |

Table 3: **Parameters we explored for the optimized baseline**. Not all combinations were tried. We tuned in order: learning rate lr, filter sizes, max-k pooling, tuning embeddings, batch size, classifier depth and lastly tried another optimizer.

| | |
|---|---|
| Filters | {1: 57, 2: 29, 3: 14}, {1: 57, 2: 29, 10: 14},{1: 285, 2: 145, 3: 70}, {1:10, 10:10, 1:10}, {1:15, 2:10, 3:5}, {1:10}, {1:100}, {10:100} |
| Filter sizes | 1, 2, 3, 10 |
| lr | 0.01, 0.0075, 0.005, 0.001, 0.0005, 0.0001 |
| bs | 1536, 4096 |
| max-k | 1, 3, 7 |
| classifier | two_layer_classifier, 'conf':[{'do': None—.2, 'out_dim': 2048 — 4196 — 1024}, {'do':None— 0.2}]}, one_layer_classifier, 'conf':[{'do':.2}]} |
| tune embedding: | True, False |
| optimizer: | ADAM, ADABOUND by Luo et al. (2019) (very low results) |

For the baseline we found optimal hyperparameters to be: lr=0.0075, filter-sizes={1: 57, 2: 29, 3: 14}, clf=one_layer_classifier, 'conf':[{'do':.2}] , max-k pooling=3, bs=1536, tune embedding=True, optimizer=ADAM with pytorch defaults. Increasing the filter size, classifier size or depth or using more k decreased dev set performance due to increased overfitting. In general the standard multi-label BCE loss overfit much more quickly than the contrastive methods described in §3. The contrastive model only differs it was able to use more filters {1: 100, 2: 100, 3: 100}, where using only {1: 20, 2: 20, 3: 20} loses 1.5 %p of performance, and that its optimal lr = 0.0005, while the batch size shrinks to 1024 due to increased memory requirements of label matching. This contrastive models optimal matcher classifier is deeper, due to the increased task complexity – four_layer_classifier, 'conf': [{'do': 0.2}, {'out_dim': 1024, 'do': 0.1}, {'out_dim': 300, 'do': None}, {'out_dim': 1, 'do': None}]}.

## A.2  LABEL-EMBEDDINGS, PRETRAINING EFFECTS ON THE LONG-TAIL

In this section we analyse the effects of using supervised label-embeddings and self-supervised pre-training with words as pseudo-labels. Plotting the average precision of 1305 would be unreadable.

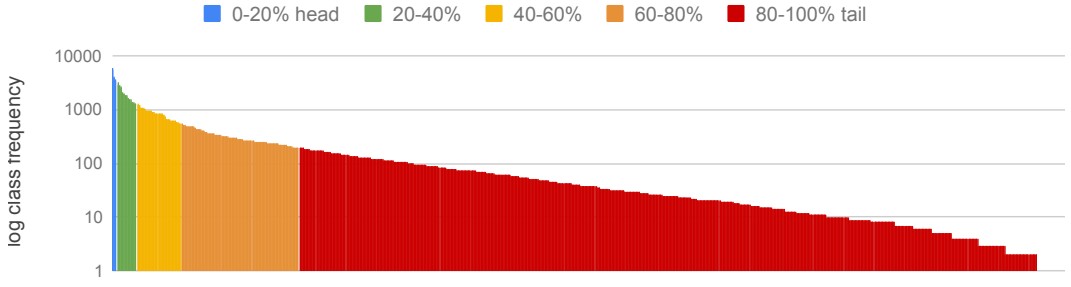

Figure 4: **Head and long-tail as 5 label frequency balanced class buckets:** We bucket classes by label frequency, into 5 buckets, so that each bucket contains equally many label occurrences – i.e. the buckets are balanced and thus comparable. Note the log frequency scale.

Instead, we sort classes from frequent to rare and assign them to one of five $20\%$ class frequency buckets, such that each bucket has the same amount of positive labels (label occurrences) in it. As seen in Fig. 4, this means that the head $0-20\%$ bucket (left, blue) has very few, frequent classes, while tail buckets $20-40\%\ldots80-100\%$ have increasingly more classes (right, orange, red) that also become increasingly more rare. We bucket classes this to balance label frequency between buckets to make them directly comparable.

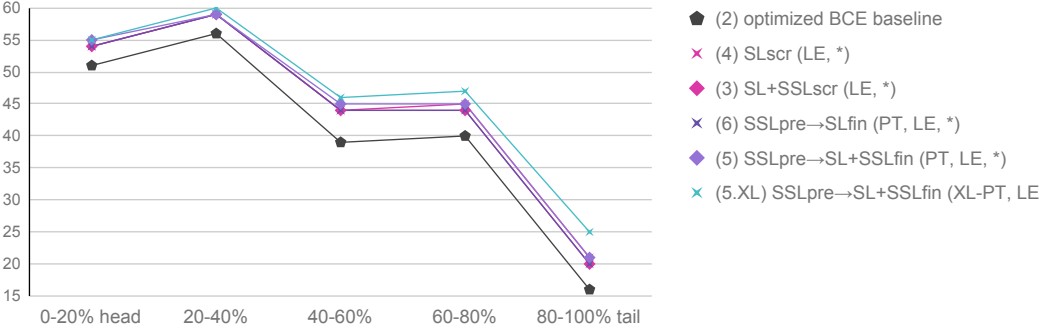

Figure 5: **Long-tail effects of base, label-embeddings and self-supervised pretraining (XL):** When reporting $AP_{macro}$ of the five $20\%$ head to tail class bins, the BCE multi-label objective performs worst. Label-embedding NCE (LE, (4-6)) markedly boosts performance, especially on the long-tail. When using label embeddings for self-supervised pretraining with the same network parameters (*) for all LE models, there is no boost. However, when pretraining with more parameters ((5.XL) – see `larger net, 3.3x labels` in the zero-shot learning Fig. 3), we see substantial long-tail performance boosts (turquoise, upper most curve).

**Label-embedding increase long-tail performance:** In Fig. 5 we can see that the optimized baseline (2) from Tab. 1 performs much worse than models that use only the supervised label-embeddings (LE) and methods that also use self-supervised pretraining (PT) via noise contrastive sampling of input words as pseudo labels. We also see that regarding end-task performance on the tag predition task, training from scratch (LE, pink $\times,\diamond$) performs only slightly worse than fine-tuning after self-supervised pretraining (purple $\times,\diamond$). However, we also see that increasing the model size during

self-supervised pretraining (5.XL) boost performance on the long-tail, especially with increasingly tailed or rare (60-100%) classes. Previously, we saw in Fig. 3 that the same "larger net, 3.3x labels" model (5.XL) increased zero-shot performance over the default parameters (*), which demonstrates that improved self-supervised zero-shot performance translates into better supervised end-task fine-tuning performance. We also found that increasing the size of non-pretrained (trained from scratch) models did not improve end-task performance, despite hyper-parameter tuning.

This leaves us with two insights for modeling. First, for small-scale, 'data-internal', self-supervised pretraining a larger pretraining model increases long-tail performance, whereas Hooker et al. (2020a) found that compressing larger models first 'forgets' long-tail performance – both experiments provide evidence that model capacity and long-tail performance are tied. This seems to be the case even for small-scale self-supervised pretraining, i.e. it demonstrates that despite training on small data, we still need increased self-supervision signals and model size to capture long-tail information, which could explain why large-scale pretraining and models perform so well rather than simply assuming them to be overparameterized. Second, this larger pretraining model has an end-task $AP_micro$ score of 49%, which is only .8 percent points better than the pretrained model (5) at 48.2% with default parameters (*), despite showing promising improvements on long-tail classes, which together with the zero and few-shot insights underlines that optimizing for learning insights and analysis other than supervised performance summary metrics can lead to a broader understanding of neural learning processes and modeling effects.

**The head is learned first (in early epochs), pretraining learns the tail much faster:** In Fig. 6 we compare early epoch training with late (optimal) epoch test scores per class bucket. We see that all models learn the head classes during the first epochs (- - dashed line). Methods (5, 5.XL) that use label-embedding (LE) and self-supervised pretraining (PT), start learning the long-tail during the first epoch, while BCE multi-label baseline (2) does not start learning the long-tail until epoch 10, and even then at a much lower performance than the pretrained label-embedding methods. Finally, we see that pretraining a larger model with more self-supervised pretraining signal (5.XL or "larger net, 3.3x labels" in Fig. 3) increases performance on the long-tail, even during the first epochs.

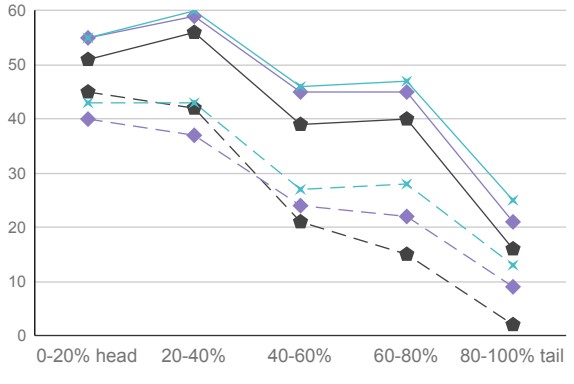

Figure 6: **The long-tail is learned during later epochs:** $AP_{macro}$ performance over five frequency balanced 20% head to tail class buckets. All methods (2-5.XL) learn the head classes ($0 - 20\%$) first – during early epochs. Self-supervised pretraining (5, 5.XL), via pseudo label-embedding NCE (LE, PT), learn the long-tail (see $60 - 100\%$) even in the first epoch (- - dashed line) of supervised learning. The baseline (2) struggles to learn this long-tail at epoch 10 and until its epoch 82 – i.e. its optimal dev set score epoch.

**Self-supervised label-embedding pretraining boosts long-tail performance:** We thus conclude, that self-supervised pretraining helps us learn the long-tail better, and faster – i.e. even after a single epoch of supervised training. This is a useful property in supervised learning scenarios where data or computation cycles are limited. **We note that: our pretraining and fine-tuning do not require**

learning rate scheduling or normalization layers like BERT or RoBERTa (Devlin et al., 2019; Wang et al., 2020c).

**Miscellaneous result/ an open future evaluation problem:** Finally, since average precision ($AP$) has no explicit notion of over and under predictions, we plotted over and under predictions per class and over all classes. In standard classification, i.e. discrete, evaluation we would have over predictions as false positives and under predictions as false negatives. Using continuous measures such as $AP$ has computational advantages and does not limit evaluation to a single threshold like $F_1$ or $accuracy$ do. However, has no notion of over and under predictions, which especially regarding long-tail issues may have a significant impact. However, plotting over and underpredictions per class (and overall) was only mildly informative. Due to the high label sparsity, we saw close to zero over predictions on average, but all-class average underpredictions were much harder to reduce (optimize). This observation is a reflection of the high label sparsity, i.e. at most $0.3$ of labels are active per instance, combined with a long-tail distribution – i.e. many rare events. Under this combination meaningful evaluation of prediction behaviour is hard to analyse in a meaningful, concise fashion, because per class plots become large and hard to fit and interpret in a paper, and all-class averages do not reveal class dynamics. We include these observations because we found them instructive to outline the challenges of evaluating long-tail learning and do not include the per class plots, because they would not be readable. We are however happy to discuss them upon request.

### A.3 FEW-SHOT: SCRATCH, PRETRAINED, ADDITIONAL SELF+SUPERVISED SCENARIOS

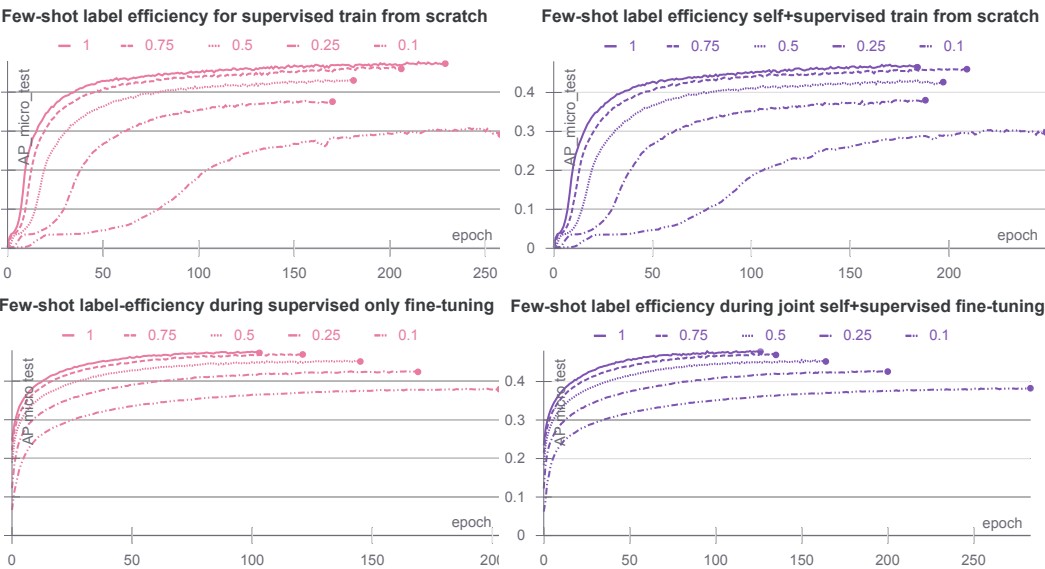

Figure 7: **Few-shot training from scratch (top 2) vs. after pretraining (bottom 2)**: and using only supervision to fit the end-task (left) vs. jointly using self+supervision (right). Results are in $AP_{micro\_test}$ for different few-shot training set portions (1, 75%, 50%, 25%, 10%). Insight 1: self-supervision during end-task fitting makes no learning difference – i.e. when comparing top (or bottom) left (supervised) vs right (self+supervised) sub-figures, they look nearly the same. Insight 2: *Pretraining (bottom figs.) via self-supervision markedly improves few-shot learning performance, speed and stability*, independent of fine-tuning via supervision (left) or self+supervised (right).

**Few-shot challenges:** Few-shot learning increases the long-tail problem. For 10% few shot learning, we train on 6800 instances, so many classes will be unseen at training time We will publish both the parsed data splits and a cleaned code version on Github to encourage experimenting with and extending to other low-resource 'dense-to-dense' self-supervision methods, additional evaluation metrics and datasets.

**Few-shot, with and without self-supervision – as pretraining or for joint self+supervised fine tuning:** Fig. 7 shows in more detail that the pretrained model (bottom) learns better, and that joint self+supervised end-task training (scratch or fine-tuned) makes no difference.

### A.4 Text preprocessing details

We decompose tags such as 'p-value' as 'p' and 'value' and split latex equations into command words, as they would otherwise create many long, unique tokens. 10 tag words are not in the input vocabulary and thus we randomly initialise their embeddings. Though we never used this information, we parsed the text and title and annotated them with 'html-like' title, paragraph and sentence delimiters. The dataset is ordered and annotated by time. Dev and test set are therefore future data compared to the training data, which results in a non-stationary problem, though we never determined to what extend.

### A.5 Potential ethical considerations

In this section, we outline potential impacts of our work for machine learning practice, as well as its possible environmental, societal, health and privacy implications. As with any technology there is the dilemma of dual use (Rappert & Selgelid, 2013). Below, we briefly discuss beneficial and potential detrimental impacts of this work as we can foresee them Hovy & Spruit (2016); Brundage et al. (2018).

The main goal of our research is to reduce the hardware and compute requirements of current representation pretraining methodology for language representations, especially for challenging low low-resource, long-tail problems. Due to the reduction in compute requirements, our methods may help reduce carbon impact and the exhaustion of precious resources like rare metal compared to large-scale pretraining. Consuming less energy and mining less resources for hardware production has major impacts on the environment (Tsing et al., 2017). Thus, as a research community we should take action not to let AI methods become a race for precious metal hardware due to its devastating effects on our shared environment. Further, small-scale pretraining could make access to modern NLP methods easier for machine learning researchers and practitioners, who have less hardware resource privileges than are required for state-of-the-art solutions, or whose language of research does not allow for easy access to Web-scale text collections. This may become even more important as socio-economic factors are likely to play a fundamental role in the future democratisation and fair access to AI technology (Riedl, 2020) for economics, health and other key decision making areas. This is especially important as large-scale hardware resources increasingly lead to research and economic inequalities as described by Hooker (2020); Riedl (2020). Another important advantage of researching more data-efficient methods is that using as little data as needed is a requirement of the GDPR regulations for 'privacy by design'.[3] This principle is in direct conflict with the current self-supervised pretraining approaches, which parties who have both access to massive data collections and compute resources predominantly study.

Furthermore, there may be potential implications in better learning of underrepresented and rare events from small or very limited data collections (Mitchell et al., 2020). When we increase self-supervision during pretraining, i.e. when pretraining on more diverse learning signals than direct supervision can provide, we see a substantial increase increase in few-shot (low-data) performance, which, upon inspection, becomes clear is caused by a better retention of rare event performance than direct supervision could provide – see Fig. 2. However, we did not yet study whether this pretraining reduces or increases unwanted data biases (Waseem et al., 2020), though typical analyses of gender and racial biases may be hard on the current dataset of machine learning questions. Note that we did not chose this data set to solve a specific application task, but only as a proxy to study the effects of small-scale pretraining on challenging data.

Better small-scale pretraining could benefit areas like medicine where large pretraining is not as effective or fails for a lack of external data resources (Şerbetci et al., 2020). Due to the usage duality of research in general, research into more resource-efficient learning could also cause privacy concerns, enabling easier surveillance, and improved advertisement recommendation can have unforeseen political, but also economical and even environmental impacts, as the goal of advertisement is increased resource consumption.

Thus, a general approach to furthering beneficial usage over detrimental applications of dual technology should regard applying ethics principles at every step of reuse of the discussed methods to

---

[3]https://en.wikipedia.org/wiki/Privacy_by_design

support its transparent use and public verification and auditing, to protect vulnerable groups from harmful applications.

