# OpenReview forum: "Self-supervised Contrastive Zero to Few-shot Learning from Small, Long-tailed Text data"
_ICLR.cc/2021/Conference — Reject_

### Official Review · AnonReviewer3 · 2020-10-25
**Focused, well-written work but limited to specific types of tasks**

**Rating:** 5
**Confidence:** 3

**Review:**

##########################################################################
Summary:
This paper studied the effect of pretraining from a miniaturization perspective. The proposed, dataset-internal contrastive self-supervised learning benefits in zero- or/and few-shot learning settings.

##########################################################################
Reasons for score:

Overall, my score is marginally below the acceptance threshold.

1. The paper provides a very in-depth survey of recent work on pretraining research and a clear scoping of the problem to be addressed in this work against the other work. I also like the framing of dataset-internal and dataset-external to distinguish between self-supervision and task-supervision, respectively. I also like the list of limitations of the prior work in the Related Work section.
2. The description of the proposed method in Section 3 was very detailed and easy to follow.
3. Overall messages and findings from the experiment were clear and convincing to me.

Cons:

1. Although the clarity of methodology description and clear motivation compared to prior work, my main concern of this work is the lack of generalizability of the proposed method to other low-resource, long-tail problems. The predicting pseudo, noisy labels itself is not a novel idea. The main novelty of the method comes from predicting new labels from word token space directly, which seems to be very limited to the task itself; free-formed tag labels for text. As described in the conclusion section, the method seems to be only applicable when input text and label tags are in the same place.

2. Another concern is the unclear description of how two binary labels are optimized in a contrastive way. Usually, there should be an additional loss to maximize the difference between two similar-looking examples of different labels in contrastive learning like SIMCLR. I only find that (5) in Figure 1 is a simple aggregation of binary classifiers, rather than contrastive optimization. When the batch set of positive and negative (pseudo) labels are prepared in L_i, does the ordering of the labels matter? Also, again do you pair positive and negative labels for pair-wise optimization?

3. I like to suggest changing the framing of this work from the general findings of pretraining schemes to task-specific findings. That is, Section 4 should be introduced earlier than Section 3, and describe how difficult the task is and how existing pretraining methods fail to achieve good performance. Until section 3, I had a hard time understanding whether the proposed method in Figure 1 is generalizable to any downstream tasks or specific to certain tasks. The partial examples in Figure 1, like “measuring an interaction”, “interaction”, “p-value” are mentioned without even describing what they are. Otherwise, it would be really nice to add more applications in a long-tail, low-data regime, proving that the proposed framework is general enough.


##########################################################################
Questions during the rebuttal period:

Please address and clarify the cons above.
Here are some questions:
1. In section 3, what do you mean by “Sampling pseudo-labels provides a straight-forward contrastive, partial autoencoding mechanism…”?
2. In Section 6.2, authors controlled the hyper-parameter for a better-controlled experiment. However, I wonder this is a fair game. Each training technique may have different levels of optimality. So, I guess given a fixed set of parameter ranges, providing the optimal setting might be a more fair setting. Or, at least providing them in the Appendix might be more convincing.

#########################################################################
Some typos and suggestions for presentation:
(1) Too many styles in the text (e.g., bold, italic, the numbers) make it difficult to read sometimes. I know they are used to help follow the main points but they are quite distracting.
(2) Table Tab. 2 -> Tab. 2

---

> ### Author Response · Authors · 2020-11-17
> **'text-to-text' task applications, clarification contribution and loss - share question answers in (GP) comment above**
>
> We thank the reviewer for the raised points, which made us aware of necessary improvements regarding contribution/novelty, optimization and general applicability of the methods. We edited the submissions related work, to better answer these questions.
>
> ### General application:
> - see general reply
> - [b] proposed modeling "any NLP task as a 'text-to-text' task". This works well with large-scale ,'data-external', self-supervised pretraining - but not for small-scale pretraining [2.1,2.2]. Our focus is on analyzing the effects of small-scale, self-supervised pretraining for an NLP 'text-to-text', but under a low data resource scenario, i.e. a small dataset, with a long-tail, because in such a long-tail data is always limited. We wanted to avoid solving a specific task (dataset), and picked an NLP dataset that is generic enough to investigate small-scale, self-supervised 'text-to-text' pretraining and has a challenging, small label set.
>
> ### Clarification contribution 1.
> - (I) Label embeddings have been used for supervised-only pretraining [a, c, d] but not yet been tied to self-supervised pretraining. (I) Self-supervised pretraining is common in 'data-external', large-scale pretraining of 'text-to-text' tasks. We combine both methodologies (I+II). (II) is typically trained using a softmax, i.e. by translating from dense input embeddings into sparse class representations. We avoid this translation in modeling NLP as a 'dense(text)-to-dense(label-text)' task instead. We train our method with noise contrastive estimation (NCE) [6] and use pseudo-label embeddings (words) for self-supervised pretraining. While using predicting words is not new, we predict word embeddings instead of label indicators. Pseudo-label embedding NCE has advantages over using label indicators via a softmax.
>     - One, labels (label-semantics) can be pretrained (word-embeddings) - label indicators can not.
>     - Two, no cumbersome dense-to-sparse encoding translation into label indicators, or worse in, language 'text-to-text' models we do sparse-to-dense-to-sparse back and forth translation.
>     - Three, NCE training with a single class matcher avoids computing a softmax - we updated the submission accordingly. It allows arbitrarily large class (and word) sets and is inherently zero-shot prediction capable, while label indicator are mutually exclusive, and have no similarity like label-embeddings. For humans, labels are represented by words with semantics, not as indicators, we reason that models label should also have semantic representations (label-embeddings) which are known before training time.
> - These reasons are why we design words and labels to share the same space. This is not a limitation because new labels or words can still be constructed, e.g. from words in context, label descriptions etc., but ideally are know through word labels.
> - Our proposal is thus: "use label-embeddings" for self-supervised, small-scale NLP pretraining instead of label indicators (used in large-scale pretraining), because label-embeddings are zero-shot capable and are known to benefit long-tail prediction. We aim to understand the effects of *increasing small scale self-supervison* on zero and few-shot learning to avoid needing massive data as in GPT3 etc. We chose model and dataset to suit this goal.
>
> ### Loss 2.
>
> - The submission now mentions NCE [6], which uses positive and negative examples with a single binary matcher class. With label-embedding it is used to compare (dis-)similar input-label pairs. Other contrastive losses compare (dis-)similar input-input pairs, which are hard to contruct in NLP - unless a corpus with similar text pairs is provided, e.g. sematic textual similarity oder paraphrasing. We arrange positive and negative labels as one of either per sample - a batch of single class predictions, hence the (Fig1) t.repeat(text-encoding) operation.
>
> ### Controlled hyperparameters
> - Controlling the hyperparameters was inspired by [e] as we wanted to know what effects self- and supervised noise contrastive training modes have on supervised performance (ablation wise). If the primary goal were to maximize supervised end-task performance, we recently found that increasing the self-supervision pretraining encoder size leads to 1% point increase in end-task performance. Should we add this result to the appendix? The other methods did not benefit note-worthily from increasing model capacity or negative sampling amount - while doubling memory requirements.
>
> ### Style
> + is reworked
>
> ### References:
> [1-7] refer to the common reply (GP)
>
> [a] "Deep learning for extreme multilabel text classification", J. Liu, - 2017
>
> [b] "Exploring the Limits of Transfer Learning with a Unified Text-to-Text Transformer", C.Raffel - 2019
>
> [c] "Multi-task label embedding for text classification", H. Zhang, - 2018.
>
> [d] "GILE: A generalized input-label embedding for text classification.", N. Pappas - 2019
>
> [e] "A metric learning reality check." K. Musgrave -  2020.

---

### Official Review · AnonReviewer1 · 2020-10-28
**Please compare your method with existing baselines**

**Rating:** 4
**Confidence:** 5

**Review:**

This paper proposes a contrastive autoencoder approach that only requires small data to perform a multi-label classification on the long-tail problem. They introduce a matching network to compare text and label embeddings and calculate the probabilities of the label given the input. The proposed idea is very straightforward by combining a matching network with contrastive learning to give broader signals. The goal of this work is to enable zero-shot and few-shot learning with very few resources as a more sustainable approach to machine learning applications.

In general, the idea is interesting, since it leverages the in-task data for distinguishing positive and negative samples. However, the work itself is not entirely new, there is already similar work on this [1]. I could not find enough related work on matching networks [1] and it seems the authors need to give more credit to other seminal work that has been used for many applications for few-shot learning [2]. I would ask the authors to add more relevant work to the paper

The structure of the paper can be further improved. The naming is not intuitive, such as $(SLpf, S(+S)Lpf)$. The results in Table 1 isn't easy to read (especially the learning setup).

Strengths:
- A straightforward method for self-supervised contrastive label-embedding prediction using in-task data.
- This paper focuses on a challenging, noisy long-tail, low-resource multi-label text prediction task

Weaknesses:
- The key concern about the paper is the lack of comparison to the baselines. Since the proposed method is not the first on this topic, the authors should compare their method with other well known baselines for few-shot learning, such as matching networks and Siamese Networks
- The paper does not analyze the effectiveness of the approach on the long-tail scenario.
- No ablation study on the contrastive learning and label embeddings

***Post-rebuttal***

> I want to thank the author for addressing my concerns. However, I still have issues with the evaluation and the clarity of the paper. I think the paper requires another round of revision before it is ready for publication.

References
1. Transferable Contrastive Network for Generalized Zero-Shot Learning 2019 ICCV https://openaccess.thecvf.com/content_ICCV_2019/papers/Jiang_Transferable_Contrastive_Network_for_Generalized_Zero-Shot_Learning_ICCV_2019_paper.pdf
2. Siamese Neural Networks for One-shot Image Recognition ICML Deep Learning Workshop 2015 https://www.cs.cmu.edu/~rsalakhu/papers/oneshot1.pdf
3. Matching Networks for One Shot Learning NeurIPS 2016 https://arxiv.org/pdf/1606.04080.pdf

---

> ### Author Response · Authors · 2020-11-17
> **Relation to vision and contrastive baselines we used from NLP + questions share between reviewer found in general reply (GR).**
>
> We thank the reviewer for the comments. We highlighted the connections to baseline methods/ architectures we used from earlier NLP more strongly and added the suggested, but newer related works from computer vision. Given the additional page, we also mention matching networks as suggested along with noise contrastive estimation as we believe both are instructive in understanding the origins of related ideas.
>
> ### Baselines:
> - The suggested vision work [c] uses the same (or very similar) matching architecture as NLP work [b] we cited in the original submission, where the NLP work is older. We use the NLP version of the architecture and its training principle as a baseline for supervised contrastive training - i.e. when not using self-supervision. However, the NLP baseline and the suggested vision analog used only *supervised pretraining for transfer or zero-shot learning*. We use fully self-supervised pretraining - i.e. by autoencoding (rematching) text words to use them as pseudo-labels. Thus, we do not use labels for pretraining or zero-shot prediction, while the (originally) cited and suggested methods do. Because of NLPs' strong reliance on (large-scale) *self-supervised* transfer learning, our evaluation setup and research focus is geared toward analyzing *self-supervised transfer rather than supervised transfer.* Since, the suggested vision reference analyzes supervised transfer for zero-shot learning, our  method and self-supervised evaluation goals are not directly comparable to the reference. However, we agree with the other references and mention them now. We originally related to many recent contrastive or metric learning works with the critical and very instructive metric learning review by [a], because it covers many vision methods and the very relevant NLP architecture (baseline) was already cited in the original submission.
>
> ### Regarding weaknesses
> - We used a baseline architecture from NLP (cited [b] in original submission) that is very similar to the one suggested from vision.
> - Ablating label embeddings and contrastive learning amounts to using multi-hot predictions with multi-label binary cross entropy, which is a baseline called BASE in the original submission. Since our intended focus is on analyzing self-supervised and zero-shot evaluation given small, challenging, data we do not ablate label embeddings, as they are required for zero-shot learning. We do not ablate contrastive learning without label embeddings, as this would require a Siamese architecture to compare positive and negative input samples. But since no two texts are generally similar enough for positive samples, siamese architectures in NLP are limited to tasks where similar text inputs are available - e.g. semantic similarity and paraphrasing. Generating such samples for any NLP task, e.g. our task or general 'text-to-text', is an open research question and an interesting idea for future research, but not an ablation or a baseline. In vision input-input contrastive learning makes sense, because there are many images with similar content. In NLP input-label contrastive learning makes sense, because texts have labels, next words, summaries etc. Vision is somewhat limited to input-input contrastive learning like SIMCLR, but can use word label embeddings to allow for input-label contrastive learning which seems to be exactly what was exploited suggested work suggested during the review as a baseline (but covered by older NLP baselines we used).
> - As suggested, we will include an analysis of how the long-tail performance is affected by the different methods in the November 24 submission because label embeddings are a common technique for long-tail problems. The experiments for creating appropriate plots are still running as we had to find an meaningful plots to visualize > 1.3k class performances, which we will include in the appendix as they foreseeably take much space. First results suggest that all long-tail performance is improve for supervised and self-supervised contrastive method.
>
> ### References:
> [1-7] refer to the common reply (GP)
>
> [a] "A metric learning reality check." Kevin Musgrave -  2020.
>
> [b] "Multi-task label embedding for text classification", Honglun Zhang, - 2018.
>
> [c] "Transferable Contrastive Network for Generalized Zero-Shot Learning", 2019

---

> > ### Author Response · Authors · 2020-11-23
> > **Long-tail analysis added**
> >
> > We added the long-tail analysis to the appendix A2. These results were insightful and we thank the reviewer for the request.
> >
> > ### Overall, there are some new take-aways from the analysis:
> > 1. All methods, either using label-embeddings NCE or multi-label BCE, learn the head (frequent) classes first -- unsurprisingly.
> > 2. The label embedding methods result in substantially better long-tail performance -- in agreement with the related research.
> > 3. Self-supervised label-embedding pretraining via NCE allows learning long-tail classes within the first epoch and overall learns it much faster than the standard multi-label approach.
> >
> > We also noticed that a larger pretrained model with more negative and positive word sampling for self-supervised pretraining learns faster, i.e. reaches its optimal few, zero- and long-tail performance in fewer epoch -- optimal according to dev set performance.
> >
> > ### Style
> > + We updated the style as requested and marked revisions in mute blue.
> > + Model setups/ experiments now use consistent numbers.
> > + As in the original these **ablate**:
> >    + self-supervised vs. supervised label embedding combinations. Pretraining is always self-supervised (for zero-shot evaluation). Fine-tuning and training from scratch can be supervised or self-supervised + supervised.
> >    + Using label-embeddings vs. no-label embeddings (BCE base model)
> >    + Other ablations are addressed in the previous comments

---

### Official Review · AnonReviewer2 · 2020-10-28
**Good motivation but experiments are limited**

**Rating:** 5
**Confidence:** 4

**Review:**


Quality


I like the intended focus of this paper which is to perform self-supervised training of small data for downstream tasks with applications for zero and few-shot learning. However, there are many limitations.


(1) The only task considered is multi-class classification. The technique proposed in this paper revolves around learning the label embeddings that would match the input embeddings, which is quite limited to multi-classification and might not carry much value to other tasks. For example, for tasks such as entailment prediction where the labels are `true' or `false', for the model to perform well, most of it comes from the ability to process the input and does not have much to do with the model's understanding of these two labels, which is unlike in multi-class classification where label understanding is crucial. Therefore, the proposed approach seems to be quite limited to the areas of multi-class classification.
(2) I would love to see more experiments on other tasks besides multi-class classification but wonders how applicable the framework will be. Explanations would be appreciated here.
(3) Learning label embeddings seems like a good idea and this seems like an ok way to do it. However, label embeddings are somewhat learned already in some pre-trained models such as generative sequence-to-sequence models (T5 or related models). The paper at least need to have these models as baselines for comparison.


Clarity

The writing is at an acceptable level. However I find it a bit dense in terms of explanations and might need more illustrative figures.

- I don't really get see the CNN architecture is used here. I have some concerns that the method might not be applicable for other architectures and would like to get some more clarity on this.
- It is not entirely clear why the prevalent evaluation favors large self-supervised pre-training. ***
- Too many notations (s, -, +s , SL) that makes it hard to follow in the results section.




Originality

- The paper seems heavily inspired by SIMCLR so in my opinion the originality is low to moderate.


Significance


The premise of this work is to avoid having to use external datasets for pre-training. However, the prevalent self-supervised training in many work (BERT, RoBerTa, to name a few) do not require labels, so I don't see why we really need to avoid that. The paper can be positioned a bit better if the paper uses these models, and performs additional learning on top with the 'dataset-internal' contrastive learning approach in order to improve performance on long-tail zero and few-shot learning.

However the paper does show that it is possible to do well on long-tail zero- and few-shot learning with only small data pretraining, so that is a nice conclusion derived from the experiments. However, I am not entirely sure about the real-world applicability.


High-level pros and cons


Pros
- Long-tail zero and few-shot learning is an interesting direction which the paper explores

Cons
- This paper seems like a probing paper that looks into how well we can do without external pre-training, but might not be so applicable for real-world applications due to reasons above (see discussion in Quality and Significance)

---

> ### Author Response · Authors · 2020-11-17
> **Reviewer specific reply, questions shared between reviewers found in general reply (GR) above**
>
> Thank you for the instructive review and questions, it helped us make our intent and research relations regarding NLP methodology more clear.
>
> ### Answering Q(3)
> We fully agree with the relation to T5s' 'text-to-text' prediction as this was exactly our intent. Instead of label indicators for text, we run 'dense(text)-to-dense(text)' prediction via noise contrastive estimation [6]. This enables infinite label (class) an word set prediction, e.g. language models predict a large vocabulary (classes), as do we during self-supervised pretraining, where we predict words from a text (postive pseudo labels) and negative words from other texts in a batch. Unfortunately, Transformer pretraining neither seems to work yet for small language model pretraining (compare [4,5]), nor do contextualized embeddings work as label embeddings (see [3]) — word embedding are still needed for long-tail BERT prediction.
>
> ### Answer: Q(1+2)
>
> This review helped us clarify the ubiquity of the method in the paper. Similar models are used in vision (reviewer 3 pointed this out) — though only for supervised pretraining/ transfer. Previous label-embedding work (see original submission) used only supervised pretraining for transfer as well. They used matching CNNs with label embeddings, as did long-tail prediction methods that do not require external pretraining [a]. As mentioned above, due to Transformers' limitations (updated submission RR section) we thus use label-embedding CNNs. We use previous works' architectures (cited in original submission) as a base(line) but  combine them. We merge the label and word embeddings, use deeper encoders, but more importantly, add words as pseudo-labels for self-supervised pretraining, giving the architecture inherent zero-shot learning capabilities. Our primary goal is to analyze the benefits of small data self-supervison (so far not researched), thus we limited ourselves to a (probing) dataset that is small enough and difficult enough within a very limited data settings, where the long-tail further limits label/word data. For larger datasets, analysis is near pointless, because many already do this research with (pretrained) transformers. We want to understand small alternatives. Further, since in NLP, words are classes, e.g. in language models or translation, we use word as pseudo-labels for self-supervised pretraining. We propose using self-supervised word embedding prediction as pretraining not self-supervised pretraining on word indicators (softmax). When labels and words are embeddings one can: pretrain labels (their semantics), handle arbitrarily large vocabularies and class sets (long-tail via NCE) and predict over unseen word/label embeddings (zero-shot) out of the box. There are many workshops, e.g., REPL4NLP, on deriving unseen word- and hence label-embeddings. We want labels and words in the same vector space to inherently enable zero- and few-shot prediction while covering semantic drifts. This is not a limitation but a design choice.
>
> ### Clarity
>
> - The CNN is used as the text encoder, as in previous work.
> - There are various critiques on how meaningful current task evaluation is: e.g. testing large scale memorization [c,2.1,2.2], random seed hacking [b,d], Clever Hans evaluation [e].
> - We will improve annotation towards the 24.
>
> ### Originality
>
> - SIMCLR is mentioned as a successful contrastive self-supervised learning method for image representations but otherwise unrelated. If NLP adapted it that would be a novel, no?
>
> ### Significane
>
> - **Clarification**: Our method *does not require labels for pretaining and does zero-shot learning without labels*, but is designed to work on very small pretraining corpora - i.e. so anyone can pretrain, e.g. in domains like non-English, NLP related domains where Transformers underperform much smaller models [f]. Big Data, external (transformer) pretraining works very well, but does not work to efficiently learn (pretrain) representations from small, challenging data (see [4] vs. [5] and [2.1, 2.2]), which is our core research interest. We thus propose an self-supervision method/ alternative for such a use case.
>
> ### References
>
> [1-7] refer to the common reply (GP)
>
> [a] "Deep learning for extreme multilabel text classification", Jingzhou Liu, - 2017
>
> [b] "The Hardware Lottery", Sara Hooker
>
> [c] "How can we accelerate progress towards human-like linguistic generalization?", Tal Linzen, 2020
>
> [d] "Fine-tuning pretrained language models: Weight initializations, data orders, and early stopping.", Jesse Dodge -, 2020
>
> [e] "Right for the wrong reasons: Diagnosing syntactic heuristics in natural language inference. Tom McCoy, 2019
>
> [f]  "EffiCare: Better Prognostic Models via Resource-Efficient Health Embeddings", Necip Oguz Serbetci - 2020

---

### Author Response · Authors · 2020-11-17
**(GR) General reply to shared questions, contribution, application, related work**

### Central questions:
We thank the reviewers for the helpful comments. An updated PDF version marks requested additions as *blue text*. Below, we address *recurring questions (italic)*, that relate to our experimental **motivations** and **contributions (bold)**. We address reviewer specific questions separately.

### Contributions and answers to shared questions - updated in submission:
We added clarifications about **contributions** and relations to works across fields to the related research section (2).

- **Motivation:** our central research question is: "Do we really need to 'self-supervisedly' pretrain on ever more massive datasets to get better zero and few-shot performance (seen in [2])? How can we 'self-supervisedly' pretrain on small, preferably challenging, tasks and still get large boosts to zero and few shot learning?" -- i.e. do we need massive data and models or can we just increase self-supervision? We added related research as follows: For large external data pretraining [1] demonstrates that increasing self-supervision boosts few-shot learning, keeping data size constant, while recent methods like XLNET, Roberta increase data size [2.1,2.2]. [3] showed that label-embedding fine-tuning over large-scale external pretraining boost long-tail prediction.
- **Contributions:** Hence, we investigate: "Does *increased self-supervision* when pretraining * only on small 'task-internal' data also boost zero-, few-shot and long-tail performance*?"
    - *Why label embedding CNNs*? As transformer pretraining underperforms CNN and LSTM models on small text datasets (compare [4] vs. [5]), we use fast, text CNNs and add self-supervised label-embeddings by sampling positive and negative input words (pseudo-label-embeddings) from mini-batch instances.
    - *Self-supervised Contrastive?* While language models predict word indicators, we predict word embeddings using noise contrastive estimation NCE as in [6] -- i.e. we express NLP as a 'dense(text-embedding)-to-dense(text-embedding)' matching task rather than 'text-to-text' task (T5 model etc). We propose *self-supervised NCE pretraining using pseudo-label embeddings (sampled words), resulting in a partial (noise contrastive) autoencoding objective*. This allows for self-supervised pretraining at small scale, is suitable for long-tail prediction and zero-shot learning without prior supervision, while previous label embedding methods (see original submission) required supervised label embedding pretraining.
    - *Why this dataset?* We use this dataset because it is small enough, long-tailed, noisy (hard for humans) and fits well into NLPs' prevalent 'text-to-text' prediction design. Our primary goal was to better understand the potential limits of small-scale pretraining, where text and data are limited -- as they always are in a long-tail. [7.2, sec 6] found that neural networks spend most of their capacity to encode the long-tail [7.1, sec. 3.2] and, when compressing NNs, lose performance on (underrepresented and minority) long-tail classes first. Thus training and evaluating with such challenging and small data and models is instructive to better understand pretraining and NNs through small scale experiments. We updated the related research accordingly (sec. 2), and mention NCE in the model section (3), while the data section (4) now has better reasoning of our data setup choice.

### Application specificity
- 'text-to-text' training is ubiquitous in NLP and we follow this paradigm, but avoiding softmax computation. Having labels and words in the same space is not a limitation, for humans labels are words. Also, labels can still be assembled from descriptions, subword averages etc.
- In the real world long-tail (rare) labels and (word) feature occurrences dominate, data is limited and zero-shot task are common. Hence, we want to evaluate self-supervised pretraining under this setting by default.

### Style
- We incorporated the requested style changes.

### Requested Long-tail analysis (LT)
- See appendix A2. Self-supervised embedding pretraining boosts LT performance -- esp. for larger pretraining with more self-supervision signals.

## References:
[1] "Self-supervised meta-learning for few-shot natural language classification tasks". T. Bansal 2020

[2.1, 2.2] "A survey on contextual embeddings", Qi Liu -2020. + "Learning and evaluating general linguistic intelligence". D. Yogatama - P. Blunsom. 2019

[3] "X-BERT: extreme multi-label text classification with using bidirectional encoder representations from transformers", W.C. Chang - 2019.

[4] "Pointer Sentinel Mixture Models". S. Merity - 2016

[5] "Transformer on a Diet", C. Wang, - 2020

[6] "Noise contrastive estimation and negative sampling for conditional models: Consistency and statistical efficiency.", Z. Ma - 2018

[7.1, 7.2] "Characterising Bias in Compressed Models", Sara Hooker - Emily Denton, 2020 (sec 3.2) + "What Do Compressed Deep Neural Networks Forget?, Sara Hooker, - 2020 (sec 6)

---

### Decision · Program_Chairs · 2021-01-07
**Final Decision**

**Decision:**

Reject

**Comment:**

The paper presents a self-supervised model based on a contrastive autoencoder that can make use of a small training set for upstream multi-label/class tasks.
Reviewers have several concerns, including the lack of comparisons and justification for the setting, as well as the potentially narrow setting. Overall, I found the paper to be borderline, the cons slightly greater than the pros, so I recommend to reject it.